# Attitudes of healthcare professionals in treatment decision-making for older adults with cancer: a scoping review protocol

India Pinker ![iD] , Sophie Pilleron

Ageing, Cancer and Disparities Research Unit, Department of Precision Health, Luxembourg Institute of Health, Strassen, Luxembourg

**Correspondence to**
India Pinker; india.pinker@lih.lu

## ABSTRACT

**Introduction** The number of older adults with cancer is increasing worldwide. These patients' unique care needs, arising from comorbidity, polypharmacy and frailty, often necessitate healthcare professionals (HCPs) to rely on their own attitudes and assumptions to a greater extent when making decisions due to limited evidence. Differences in patient and HCP attitudes can impact treatment decisions and patient outcomes. There is limited research, however, on HCP attitudes in treatment decision-making for older adults with cancer. This scoping review aims to explore the attitudes of HCPs in treatment decision-making for older adults with cancer.

**Methods and analysis** The electronic databases PubMed, Elsevier Embase, Medline (from Embase) and EBSCO CINAHL Complete will be searched on 4 July 2023 to identify eligible studies based on the developed inclusion and exclusion criteria. No restrictions on study period, geography or language will be applied. Screening and data extraction will be completed independently by teams of two reviewers, with conflicts resolved by a third reviewer. The review findings will be presented as tables and in a narrative summary.

This scoping review follows the framework of Arksey and O'Malley with the Levac extension. Data extraction and analysis will be performed to identify patterns and gaps in the literature to provide an overview of the attitudes of HCPs in treatment decision-making for older adults with cancer.

**Ethics and dissemination** No ethical approval is needed. The findings will be published in a peer-reviewed journal and presented at conferences, providing insights to improve treatment decision-making for older adults with cancer and guide future interventions for HCPs in geriatric oncology.

**Trial registration number** Registered on Open Science Framework at https://doi.org/10.17605/OSF.IO/T7FD3.

## INTRODUCTION

Reflecting population growth, the number of older adults within the global population is steadily rising. This demographic trend is also reflected in cancer incidence with half of these cases diagnosed in patients over the age of 65.[1] Older adults with cancer can have unique care management needs because of comorbidity, polypharmacy and frailty.[2 3] Further complicating matters, older

## STRENGTHS AND LIMITATIONS OF THIS STUDY

⇒ The search strategy for this scoping review was devised by a research librarian, enhancing the robustness of the search.
⇒ The search was not restricted in terms of language, geography or time, resulting in a more comprehensive search.
⇒ A limitation of the study, however, is the difficulty in measuring and defining the primary target outcome of attitude, potentially leading to variations in interpretation.
⇒ As it is a scoping review there is no assessment of the quality of the papers included.

adults are a notably heterogeneous group with regard to health and fitness and are under-represented in clinical trials, limiting the evidence base from which healthcare professionals (HCPs) can draw treatment decisions.[4] Consequently, HCPs may rely on their attitudes, preferences and assumptions for cancer treatment decisions in older populations. Attitudes are ascribed a functional role in guiding behaviour, particularly when confronted with decision-making under conditions of uncertainty.[5] Reaching agreement on treatment decisions under these conditions may, therefore, be more complex when attitudes differ between HCPs and patients and could directly impact treatment and subsequent health outcomes.[6 7]

Attitudes are reflected in an individual's preferences, inclinations and tendencies.[5] When patient and HCP attitudes and consequent goals differ, this can impact patient representation in treatment decision-making.[8] The significance of this issue may be heightened in cancer cases, where decisions frequently involve multidisciplinary teams without direct patient involvement. For these cases, HCPs serve as proxies for the patient, emphasising the importance of aligning the understanding of both HCP and patient preferences. If there are discrepancies

in priorities between the two, health management decisions may fail to adequately reflect the holistic needs of the patient.[9] This, in turn, could ultimately impact the cancer outcomes of the patient as they may not be as invested or adhere to treatment that does not represent their needs or preferences.[10]

HCPs often lack sufficient training that fosters a deliberate exploration of older adults' preferences during the decision-making phase of treatment.[11] This may lead to the tendency of assuming patient values based on the HCP's personal attitude—informed by their preferences, experiences and values—which may not accurately reflect patient perspectives.[5 12]

While most studies and recent reviews have focused on older adults' preferences or patient-HCP comparisons,[6 9 10 12 13] there is limited research on oncology-based HCP preferences alone.

As our understanding of the older adults with cancer's preferences develops, it is important to clarify the HCP perspective to ensure that the treatment decision process—which is often different for older adults compared with younger adults[14 15]—is not inhibited by incompatible goals. Consequently, gaining an understanding of the patterns within these attitudes is crucial to identify strategies that can effectively engage HCPs and ultimately benefit older adults with cancer.

## OBJECTIVES

This scoping review aims to provide an overview of the attitudes of HCPs in treatment decision-making related to older adults with cancer.

Specifically:
► Identify patterns in attitudes of HCPs specific to treatment decision-making for older adults with cancer.
► Identify potential contributing factors to these attitudes.
► Highlight gaps in the literature.

## METHODS AND ANALYSIS

This protocol is written based on Preferred Reporting Items for Systematic Reviews and Meta-Analyses extension for Scoping Review (PRISMA-ScR) guidelines for Scoping reviews.[16]

### Design

A scoping review will be conducted based on the Arksey and O'Malley methodological framework with the extension developed by Levac and colleagues.[17 18] The reporting of the review will be based on PRISMA-ScR guidelines for Scoping reviews.[16] A preliminary search—conducted in May 2023—of PubMed, the Cochrane Database of Systematic Reviews, PROSPERO and Open Science Framework was conducted and no current or underway systematic reviews or scoping reviews on the topic were identified.

### Patient and public involvement

No patient involved.

### Eligibility criteria

The eligibility criteria was formulated based on the Participants, Intervention, Comparator, Outcome framework[19] (see table 1).

### Information resources

We will search for papers from the following databases:
► PubMed.
► Elsevier Embase.
► Medline (from Embase).
► EBSCO CINAHL Complete.

### Search

The search strategy was developed by a health focused research librarian (table 2). Comprehensive search strategies will be employed in each of the databases, using a combination of interface command language and text keywords. The search will be formulated in PubMed, from which the strategy will be translated into other databases reflecting their command languages if this is applicable. The search strategy will be made using Boolean operators and truncation. All searches will be reported within PRISMA-ScR guidelines.[16]

Due to the limited number of hits from the above search, no search terms for the key concept of 'attitudes' were included.

### Study selection

The selection of studies will follow a multi-stage process:
► *Screening by title and abstract*: prior to primary screening, duplicate records will be removed. Titles will be screened by independent review done in a reviewer pair. Disagreements regarding the inclusion of papers for full text review will be resolved by a third reviewer.
► *Screening by full text*: full texts of all papers included after the title and abstract screening will be acquired and assessed by the reviewer pair independently, judging whether the papers meet the inclusion criteria. Disagreements will be resolved through the same process as the title and abstract screening.
► *Screening by references*: the reference list of all included full papers will be screened by the reviewers.

Screening will be streamlined by using a specific review software, Rayyan.[20]

### Data management (included and excluded papers)

All papers considered for full text screening during the abstract and title screening process that are then excluded during the full text screening will be tabulated on a digital spreadsheet with a justification for exclusion based on the inclusion and exclusion criteria. Endnote20 will be used for reference management and bibliography generation.

### Abstract-only cases

In cases where only the abstract of a potentially eligible paper is available, the authors of this paper will be contacted inquiring whether a full version of the manuscript has been published elsewhere. If it has not been published elsewhere, the study will be excluded. The

**Table 1** Eligibility criteria based on the PICO framework

| Element | Inclusion criteria | Exclusion criteria |
|---|---|---|
| **P**articipants | Papers involving healthcare professionals (eg, oncologists, nurses, physicians, nurse practitioners, etc) of older adults (mean/median ≥65 years old) with cancer, any HCP affiliated with cancer care or treatment decisions in cancer care of older adults. The disease focus is on cancer with no restriction on cancer type | HCPs in the study are exclusively associated with patients younger than 65 years old (eg, paediatric, patient-HCP comparative studies where mean/median patient age <65). HCPs or patients in studies related to diseases other than cancer |
| **I**ntervention or exposure of interest | The primary focus of the studies must be on the HCP perspective of treatment decision-making in the field of geriatric oncology. Furthermore, the outcomes of the studies should include information regarding subjective inclinations, rankings and perspectives that influence the choices made within the decision-making process. This will be indicative of HCP preferences, attitudes and their individual perspectives | |
| **C**omparator | Different HCPs (eg, oncologists and specialist nurses). Or different age groups—but this must include a group where the adults are ≥65 years old. Comparisons between patient and HCP priorities, only HCP priorities will be extracted from these studies. Comparisons are not mandatory for inclusion | None |
| **O**utcome | Any outcomes related to individual professional factors within treatment decision-making in geriatric oncology such as attitudes, perceptions, priorities or preferences. For the purposes of this review, 'attitudes' refers to any subjective inclinations, rankings, and perspectives that influence the choices made when determining the most appropriate course of treatment for their older patients. This description may be encompassed by terms such as perceptions, priorities or preferences | |
| Study design | Any quantitative design. Any qualitative design. Mixed methods designs. All types of reviews (ie, literature, narrative, systematic, etc) to synthesise commonly identified inclinations of HCPs during the treatment decision-making process | Case reports |
| Article type | Original peer-reviewed research articles | Perspective or opinion pieces, book chapters, dissertations, editorials, protocols, conference abstracts |
| Language | No restriction* | |
| Study period | No restriction | |
| Geography | No restriction | |

*Authors' institution hosts diverse language expertise to support this criteria.
HCPs, healthcare professionals.

time limit placed on this correspondence will be 2 weeks. Following this, one follow-up email will be sent with the same correspondence time limit. If there is no response, the paper will be excluded.

### Data collection process and extraction
Software will be used to compile the reviewer team's extracted data. For every included paper, two reviewers will independently extract the following data as a minimum:
► Author(s)
► Year of publication
► Study objective
► Study design (eg, qualitative, mixed methods, longitudinal, etc)
► Country of study
► Sampling method (eg, convenience, snowballing, etc)
► Sample size
► Sample description (sex/gender, age, health profession, etc)
► Description of attitudes if provided
► Descriptions of factors accounted for in the decision-making process if provided.

**Table 2** Key concepts and preliminary search terms

| Key concept | Search terms | MeSH terms |
|---|---|---|
| Healthcare professionals | "healthcare professional*" OR "health care professional*" OR "health* professional*" OR "health profession*" | Health Personnel |
| Treatment decision making | "decision making", "treatment decision", "decision making" AND treatment | Decision Making, Clinical Decision-Making |
| Older adults | Aged, "Older adult*", Elderly, Geriatric, Nonagenarian, Octogenarian, Centenarian | Aged |
| Cancer | Cancer*, neoplasm*, tumour*, tumor*, Neoplasia*, Malignanc*, *carcinom*, sarcom* | Neoplasms |

► Analytical approach used (eg, statistical modelling, thematic analysis, etc)
► Summary of main findings

For disagreement over any data point, the same review process followed during screening will be employed.

All collected data will be tabulated for presentation in the written report, accompanied by a narrative summary relating the results to provide an overview of the attitudes of HCP in treatment decision-making related to older adults with cancer.

### Assessment of quality

No quality assessment for included papers will be conducted, as suggested by both the Arksey and O'Malley Framework and Joanna Briggs Institute.[17 21]

### IMPLICATIONS

Findings of this review will provide insight into areas of focus to improve the treatment decision-making process for older adults with cancer, including variables that may be valuable in the design of future interventions for HCPs involved in geriatric oncology.

### ETHICS AND DISSEMINATION

No ethical approval is required for this research as it involves the analysis of secondary data. The results of this review will be published in a peer-reviewed journal and presented at conferences.

**Contributors** The scoping review protocol was written up by IP and reviewed by SP.

**Funding** This work is supported by the Luxembourg National Research Fund (FNR), Project n°16731054. The funder did not play a role in the write up of this protocol.

**Competing interests** None declared.

**Patient and public involvement** Patients and/or the public were not involved in the design, or conduct, or reporting, or dissemination plans of this research.

**Patient consent for publication** Not applicable.

**Provenance and peer review** Not commissioned; externally peer reviewed.

**ORCID iD**
India Pinker http://orcid.org/0000-0002-1816-050X

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
