## [Reviewer comments · BMJ Open]

ARTICLE DETAILS

TITLE (PROVISIONAL)	The attitudes of Healthcare Professionals in treatment decision making for older adults with cancer: A scoping review protocol
AUTHORS	Pinker, India; Pilleron, Sophie

VERSION 1 – REVIEW

REVIEWER	Ira Suarilah Universitas Airlangga
REVIEW RETURNED	04-Sep-2023

GENERAL COMMENTS	Comments to the Authors “The attitudes of Healthcare Professionals in treatment decision making for older adults with cancer: A scoping review protocol” Dear Authors, It is an interesting scoping review protocol exploring the attitudes of healthcare professionals in treatment decision making for older adults with cancer. Overall, everything is well written. However, more limitations are reported. Hereby, I explain my concerns in detail as written below. Abstract: Page 3, line 27: The sentence does not represent the whole population globally. I would like to ask Authors to delete or change. Page 3, line 28: Please, add the unique care needs specific to older adults with cancer which cannot find in other populations. Page 3, line 38: Please consider the researchers have sufficient language experts. Introduction Page 4, line 61-62 I don't think so the sentence supports the problem that the Authors would like to explore in this Scoping review. Please add more evidence to support the urgency of conducting review on attitude of healthcare professionals in treatment decision making in a specific age group: older adults with cancer. Page 4, line 83 Please add reference to support the evidence
---

	Method Concerning no language limitation was made, Authors should consider the project is supported by language(s) experts. Do the authors focused on general cancer or diseases/illness associated cancer?
--	---

REVIEWER	Anahat Kaur Albert Einstein College of Medicine
REVIEW RETURNED	09-Sep-2023

GENERAL COMMENTS	Authors have proposed an interesting project to explore attitudes of Healthcare Professionals in treatment decision making for older adults with cancer. Protocol appears to be appropriate for conduct of a scoping review. Limitation of this study due to difficulty in measuring and defining the primary target outcome of "attitude" could make interpretation of findings difficult but should not preclude the continuation of this project. Would be interesting to see if any congruent findings re: factors affecting HCP attitude are obtained after review of included studies.
---

VERSION 1 – AUTHOR RESPONSE

Reviewer: 1

“Dear Authors,

It is an interesting scoping review protocol exploring the attitudes of healthcare professionals in treatment decision making for older adults with cancer. Overall, everything is well written. However, more limitations are reported.

Hereby, I explain my concerns in detail as written below.”

• Abstract:

• Page 3, line 27: The sentence does not represent the whole population globally. I would like to ask Authors to delete or change.

Our response: We have duly adjusted the sentence as per your suggestion. We replaced:

“The global population is ageing, and the number of older adults with cancer is increasing worldwide.” with

“The number of older adults with cancer is increasing worldwide.”

• Page 3, line 28: Please, add the unique care needs specific to older adults with cancer which cannot find in other populations.

Our response: We have provided further detail in response to your comment. We replaced:

“These patients have unique care needs, and healthcare professionals (HCPs) often have to rely on their own attitudes and assumptions to make a decision due to limited evidence”

With

“These patients' unique care needs, arising from comorbidity, polypharmacy, and frailty, often necessitate healthcare professionals (HCPs) to rely on their own attitudes and assumptions to a greater extent when making decisions due to limited evidence.”

• Page 3, line 38: Please consider the researchers have sufficient language experts.

Our response: We thank you for your comment, this is an important concern. We wish to clarify that we are affiliated with an international institution possessing substantial language proficiency, enabling us to remove the previously mentioned restriction. This is now highlighted in the method section of the protocol, as described below.

• Introduction

• Page 4, line 61-62: I don't think so the sentence supports the problem that the Authors would like to explore in this Scoping review. Please add more evidence to support the urgency of conducting review on attitude of healthcare professionals in treatment decision making in a specific age group: older adults with cancer.

Our response: Thank you for this comment. We have provide the initial two lines of the introduction to better elucidate the growing patient population affected by HCP professional attitudes. We have, however, enhanced the first paragraph of the introduction to underscore the significance of our review with greater clarity and supportive evidence. We hope this is acceptable.

We have added “for cancer treatment decisions in older populations” to “Consequently, HCPs may rely on their attitudes, preferences and assumptions” and provided further evidence of importance to the final sentence which now reads as follows:

“Reaching agreement on treatment decisions under these conditions may, therefore, be more complex when attitudes differ between HCPs and patients and could directly impact treatment and subsequent health outcomes (6, 7).”

- Page 4, line 83: Please add reference to support the evidence

Our response: Thank you for highlighting this, the relevant reference has been provided.

- Method

- Concerning no language limitation was made, Authors should consider the project is supported by language(s) experts.

Our response: As highlighted above, our institution provides a breadth of language expertise to support our review. We had added “*Authors’ institution hosts diverse language expertise to support this criteria” to the methods

- Do the authors focused on general cancer or diseases/illness associated cancer?

Our response: Thank you for your comment. We have further specified the disease focus of the review to encompass only cancer(s) in our inclusion criteria.

In the inclusion criteria we have added: “The disease focus is on cancer with no restriction on cancer type.” Under exclusion criteria we have specified: “HCPs or patients in studies related to diseases other than cancer.”

Reviewer: 2

Dr. Anahat Kaur, Albert Einstein College of Medicine

Comments to the Author:

Authors have proposed an interesting project to explore attitudes of Healthcare Professionals in treatment decision making for older adults with cancer. Protocol appears to be appropriate for conduct of a scoping review. Limitation of this study due to difficulty in measuring and defining the primary target outcome of "attitude" could make interpretation of findings difficult but should not preclude the continuation of this project. Would be interesting to see if any congruent findings re: factors affecting HCP attitude are obtained after review of included studies.

Our response: We thank you for your insight and considerate comment, it is appreciated.